# Comparative genomics of high grade neuroendocrine carcinoma of the cervix

R. Tyler Hillman[1,2], Robert Cardnell[3], Junya Fujimoto[4], Won-Chul Lee[2,3], Jianjun Zhang[2,3], Lauren A. Byers[2], Preetha Ramalingam[4], Mario Leitao[5], Elizabeth Swisher[6], P. Andrew Futreal[2], Michael Frumovitz[1]*

1 Department of Gynecologic Oncology & Reproductive Medicine, The University of Texas MD Anderson Cancer Center, Houston, TX, United States of America, 2 Department of Genomic Medicine, The University of Texas MD Anderson Cancer Center, Houston, TX, United States of America, 3 Department of Thoracic/ Head & Neck Oncology, The University of Texas MD Anderson Cancer Center, Houston, TX, United States of America, 4 Department of Pathology, The University of Texas MD Anderson Cancer Center, Houston, TX, United States of America, 5 Department of Surgery, Memorial Sloan Kettering Cancer Center, Medical Center, New York, NY, United States of America, 6 Department of Gynecologic Oncology, University of Washington Medical Center, Seattle, WA, United States of America

* mfrumovitz@mdanderson.org

**Data Availability Statement:** The whole exome sequencing data related to this study have been deposited with the European Genome-phenome Archive (EGA) under access code EGAS00001003142.

## Abstract

In order to improve treatment selection for high grade neuroendocrine carcinomas of the cervix (NECC), we performed a comparative genomic analysis between this rare tumor type and other cervical cancer types, as well as extra-cervical neuroendocrine small cell carcinomas of the lung and bladder. We performed whole exome sequencing on fresh-frozen tissue from 15 NECCs and matched normal tissue. We then identified mutations and copy number variants using standard analysis pipelines. Published mutation tables from cervical cancers and extra-cervical small cell carcinomas were used for comparative analysis. Descriptive statistical methods were used and a two-sided threshold of P < .05 was used for significance. In the NECC cohort, we detected a median of 1.7 somatic mutations per megabase (range 1.0–20.9). *PIK3CA* p.E545K mutations were the most frequency observed oncogenic mutation (4/15 tumors, 27%). Activating MAPK pathway mutations in *KRAS* (p.G12D) and *GNAS* (p.R201C) co-occurred in two tumors (13%). In total we identified PI3-kinase or MAPK pathway activating mutations in 67% of NECC. When compared to NECC, lung and bladder small cell carcinomas exhibited a statistically significant higher rate of coding mutations (P < .001 for lung; P = .001 for bladder). Mutation of *TP53* was uncommon in NECC (13%) and was more frequent in both lung (103 of 110 tumors [94%], P < .001) and bladder (18 of 19 tumors [95%], P < .001) small cell carcinoma. These comparative genomics data suggest that NECC may be genetically more similar to common cervical cancer subtypes than to extra-cervical small cell neuroendocrine carcinomas of the lung and bladder. These results may have implications for the selection of cytotoxic and targeted therapy regimens for this rare disease.

**Funding:** The University of Texas MD Anderson Cancer Center Multidisciplinary Gynecologic Cancer Tumor Bank is supported in part by NIH P50CA83639 SPORE in Ovarian Cancer and RTH is supported by an NIH T32 training grant (CA101642).

**Competing interests:** Dr. Byers reports grants and personal fees from AbbVie, grants and personal fees from AstraZeneca, grants and personal fees from GenMab, grants from Tolero Pharmaceuticals, grants from Sierra Oncology, personal fees from BergenBio, personal fees from Pharma Mar, SA, all of which are outside the submitted work. The remaining authors report no conflicts of interest related to this work. This does not alter our adherence to all the PLOS ONE policies on sharing data and materials.

## Introduction

High grade neuroendocrine cervical carcinoma (NECC) is a rare malignancy accounting for less than 1% of cervical cancers. The more common cervical cancer types including squamous cell carcinoma of the cervix (SCC), endocervical adenocarcinoma (ACC) and adenosquamous carcinoma typically spread in a predictable fashion by direct extension into adjacent pelvic structures. In contrast, NECC often exhibits early lymphatic or hematogenous spread, leading to high rates of distant metastases detected at the time of diagnosis. As a result of its aggressive clinical behavior, epidemiologic studies have shown significantly worse overall survival for patients with NECC compared to stage-matched patients with other types of cervical cancer [1,2].

The treatment of NECC has been influenced by current practices in the management of the more common cervical cancer histologies, as well as by treatment approaches to pulmonary small cell neuroendocrine carcinomas [3,4]. Small cell neuroendocrine malignancies can arise at many anatomic sites, including small cell neuroendocrine carcinomas of the lung (SCLC) and bladder (SCCB). These extra-cervical small cell carcinomas share features with NECC including small cell diameter, high nuclear/cytoplasmic ratio, and frequent necrosis. For the treatment of SCLC a combination of cisplatin and etoposide (EP) has emerged as the chemotherapy regimen of choice based on prospective randomized trials demonstrating a superior toxicity profile when compared to older regimens [5–7]. Although EP is frequently used for NECC, it is not known if this regimen is superior to alternative platinum-based regimens used in the treatment of advanced cervical cancer of other histologic subtypes [4].

The limited NECC genomic data available to date suggest a heterogeneous molecular pathogenesis variably shaped by activating PI3-kinase or MAPK pathway mutations [8–10]. A previous report by our group described mutations detected among 44 NECCs using a targeted ~50-gene next-generation sequencing panel. In this study *PIK3CA* was identified as the most frequently mutated gene in NECC (18%), followed by *KRAS* (14%) and *TP53* (11%) [8]. A subsequent independent investigation of 10 NECC tumors also found frequent *PIK3CA*, *TP53*, and MAPK pathway mutations in NECC using cancer gene panel sequencing [9]. The only group to apply whole exome sequencing (WES) to NECC analyzed five matched formalin-fixed paraffin-embedded (FFPE) tumor samples, identifying recurrent *ATRX* and *ERBB4* mutations [10]. Of the reported *ATRX* mutations, only one (p.R250X) was predicted to result in loss-of-function, and thus the frequency of *ATRX* inactivation as a driver event in NECC remains an open question. Notably, no PI3-kinase or MAPK pathway mutations were identified in the five tumors analyzed in that study.

No formal comparative genomic analysis of NECC to other cervical cancer subtypes or to small cell neuroendocrine carcinomas from extra-cervical sites has been reported to date and it remains unknown to what extent these tumors resemble extra-cervical small cell malignancies at a genetic level. Through an ongoing multi-institutional collaboration, we performed WES on cryopreserved tumor tissue and matched normal tissue from a cohort of histologically confirmed NECC tumors. In addition to describing the mutational landscape of this rare tumor type, we also performed a comparative genetic analysis between NECC, SCLC, SCCB, and non-NECC cervical carcinomas with a goal of identifying underlying similarities that may guide the selection of chemotherapy or targeted therapies in the treatment of NECC.

## Methods

### Patients and samples

Tissue samples analyzed in this study were obtained from tissue banks at M.D. Anderson Cancer Center, the University of Washington Medical Center, and Memorial Sloan-Kettering

Cancer Center. Written informed consent for tissue banking and analysis was provided by patients at the appropriate institution, and sequencing was performed at the Cancer Genomics Laboratory at MDACC. This study was approved by the institutional review board at M.D. Anderson Cancer Center (protocols PA16-0891 and LAB02-188). Cryopreserved tumor tissue samples and matched cryopreserved normal tissue or peripheral blood samples were obtained. For DNA isolation from tumors, samples were cryosectioned and two adjacent sections underwent H&E staining. These adjacent sections were reviewed by a pathologist (J.F.) to confirm the histopathologic diagnosis and assess tumor cell content. Only tumor samples with an estimated tumor cell content $\geq$20% in at least one adjacent section without significant necrosis were analyzed. Clinical data was obtained from the medical record.

## Whole exome sequencing

Whole exome sequencing was performed as has been previously described [11]. Briefly, DNA was submitted for 76 bp short-read whole exome sequencing on Illumina HiSeq 2000 (Illumina) with a target coverage was 200X for tumor samples and 100X for matched normals. Reads were then processed using a standard workflow, as has been previously described [11]. For one sample without a matched normal (NECC017), a bespoke "common normal" BAM file was used comprised of down-sampled paired-end WES reads derived from peripheral blood from 5 donors, as has been previously described [11]. Tumor purity and ploidy were estimated from B allele frequencies and copy number profiles using Sequenza [12].

**Copy-number analysis.** Copy number variant (CNV) data were derived from WES data using a bespoke R package, as previously described [11,13]. Briefly, recurrent arm-level and focal CNVs were identified with GISTIC2.0 [14], using a log2 CN ratio threshold of $<$-.3 or $>$.3 for deletions and amplifications, respectively. A CNV was determined to be arm-level if it accounted for at least 70% of segments for a particular chromosomal arm. A q-value threshold of $<$ .05 was used. Fraction genome altered was determined as the total length of segments with log2 CN ratio threshold of $<$-.3 or $>$.3 for deletions and amplifications, respectively, as a fraction of total genome size.

**Detection of HPV integration sites.** In order to detect HPV sequences in WES data, we first constructed a custom reference genome consisting of hg19 plus the NCBI reference sequence corresponding to HPV16, HPV18, HPV31, HPV33, HPV45, HPV51, and HPV52. Paired reads were then re-aligned to this custom reference genome using the aforementioned methods. MACS2 [15] was used to identify peaks of aligned reads corresponding to HPV reference sequences. In order to predict intragenic HPV integration events, we ran three structural variation detection tools (DELLY [16], LUMPY [17], and BreakDancer [18]) on the BAM files containing reads aligned to the combined hg19/HPV reference genome. A putative HPV integration site was called if at least two of the three structural variation detection tools independently identified genomic HPV fusion events in the same chromosomal location.

**Data from other tumor types.** For the comparative genomics analysis, published mutation lists were obtained from individual sequencing projects including those examining small cell lung carcinoma (SCLC) [19], small cell carcinoma of the bladder (SCCB) [20], and non-small cell cervical carcinoma including squamous cell carcinoma of the cervix (SCC) and adenocarcinoma of the cervix (ACC) [21]. In order to facilitate comparisons between these datasets, only coding mutations on autosomal or X chromosomes were included.

Whole genome sequencing mutation data for SCLC (N = 110) were obtained from the MAF file published by George, et al [19]. Mutation data for SCCB whole exomes (N = 17) and whole genomes (N = 2) were obtained from the MAF file published by Chang et al [20]. Whole exome sequencing mutation data for cervical squamous cell carcinoma (N = 158) and

endocervical adenocarcinoma (N = 26) were obtained from the MAF file published by The Cancer Genome Atlas sequencing project [21]. The fraction of early stage I-II tumors at initial diagnosis in the endocervical adenocarcinoma/squamous cell carcinoma data sets was 77%, similar to the 73% rate of early stage I-II tumors in the NECC cohort. Due to low numbers in rare histopathologic categories, mutations from endometrioid adenocarcinoma of endocervix (N = 2), adenosquamous carcinoma of the cervix (N = 4), and mucinous adenocarcinoma of endocervical type (N = 4) were not included in the analysis.

**Mutational signature analysis.** Mutational signatures were derived using the adjacent tri-nucleotide context for each somatic single-nucleotide variant (SNV), as has been previously described [22,23]. For whole genomes (SCLC: N = 110; SCCB: N = 2) only coding SNVs were included in order to maintain uniformity in trinucleotide composition across datasets. For each SNV, the tri-nucleotide context was first identified using the hg19 reference genome. For each set of SNVs corresponding to a single tumor sample, the YAPSA R package [24] was used to estimate the relative contribution of each of the annotated COSMIC mutational signatures using a minimum signature contribution of 3% across all samples [22]. Unsupervised hierarchical clustering of tumor samples was performing with a Euclidean distance metric based on the relative mutational signature exposure, using the ComplexHeatmap [25] R package interface provided by YAPSA [24].

## Statistical analyses

Categorical comparisons were performed using a Fisher's exact test and comparisons between continuous variables were done using a Wilcoxon rank-sum test. All statistical comparisons were two-sided and a P value < .05 was considered significant. All statistical analyses were performed using the R statistical platform (version 3.3.1) [26].

## Data availability

The whole exome sequencing data related to this study have been deposited with the European Genome-phenome Archive (EGA) under access code EGAS00001003142.

## Results

### Mutation analysis

Clinical and demographic characteristics of this NECC cohort are shown in Table 1. We performed WES on 15 cryopreserved NECC tumor samples at 200X target coverage (mean 212X; range 138-295X) and also performed WES on 14 available matched normal samples (S1 Table). Among the tumors with an available matched normal sample, we detected 4,253 total SNVs and indels, corresponding to a median of 1.7 somatic mutations per megabase (Mb) (range 1.0–20.9). One tumor (NECC013) exhibited a somatic mutation rate more than ten times the median for the cohort (Fig 1A), and this tumor also contained a pathogenic *MSH2* missense mutation (p.G164R) suggesting that defective DNA mismatch repair (MMR) may explain the hypermutation phenotype observed in this tumor sample.

We next annotated potential NECC driver genes by identifying those genes contained within the COSMIC Cancer Gene Census (Tier 1) that also were found to have non-silent mutations in at least two tumor samples in this cohort (Fig 1B). Using these criteria, activating helical-domain *PIK3CA* c.1633G>A (p.E545K; COSM763) mutations were the most frequency observed oncogenic mutation in this cohort (4/15 tumors, 27%). Activating MAPK pathway mutations including *KRAS* c.35G>A (p.G12D; COSM521) and *GNAS* c.601C>T (p. R201C; COSM27887) were identified in two tumor samples (2/15, 13%), with these mutations

**Table 1. Clinical characteristics of high grade neuroendocrine carcinoma of the cervix cohort.**

| | |
|---|---|
| **Age at Diagnosis** | 37 (22–63) |
| **Clinical Stage at Diagnosis (FIGO 2009)** | |
| Stage I-II | 11 (73%) |
| Stage III-IV | 4 (27%) |
| **Mixed Histology** | |
| Small cell neuroendocrine carcinoma | 13 (86%) |
| Focal Squamous Differentiation | 1 (7%) |
| Large cell neuroendocrine carcinoma | 1 (7%) |
| **Distant Metastases at Diagnosis** | |
| No | 4 (27%) |
| Yes | 11 (73%) |
| **Contributing Institution** | |
| M.D. Anderson Cancer Center | 8 (53%) |
| University of Washington Medical Center | 4 (27%) |
| Memorial Sloan-Kettering Cancer Center | 3 (20%) |

Data are mean (minimum–maximum) or n (%) unless otherwise specified.

FIGO, International Federation of Gynaecology and Obstetrics

co-occurring in both tumors (NECC015, NECC012). Non-silent *TP53* mutations were identified in two tumors (2/15, 13%) including a predicted pathogenic missense mutation c.524G>A (p.R175H; COSM10648) and a nonsense mutation c.916C>T (p.R306*; COSM10663).

Examination of focal copy number variants (CNVs) using GISTIC2.0 [14] identified loss of 2q13 as a statistically significant recurrent event in NECC (q < .1), occurring in two tumors (2/15, 13%) (Fig 1C). Gene level analysis identified loss of the genomic region containing *PTEN* in 5 samples (5/15, 33%), and loss of *PTEN* was mutually exclusive with *PIK3CA* activating SNVs. In total we identified PI3-kinase or MAPK activating mutations in 67% of tumors (Fig 1B), including *PIK3CA* activating mutations, *KRAS/GNAS* activating mutations, and *PTEN* loss.

## HPV integration events

Using WES data to detect HPV integration events, we identified HPV16 in 2 tumor samples (13%) and HPV18 in 6 samples (40%) with a total of 53% of NECC samples having detectable HPV integration. We did not identify any NECC that were positive for multiple HPV subtypes. No relationship was observed between HPV integration events and the fraction of the genome altered by focal CN events (Fig 2A). In contrast, HPV integration was more often detected in aneuploidy tumors (N>2.5) compared to euploid tumors (Fig 2B).

We were able to map the HPV integration site with high confidence in 6 of the 8 (75%) of the tumor samples for which HPV genomic sequence could be identified in WES data. Of the mapped HPV integration sites, we found that 4 of 6 were located in the 8q24.21 chromosomal region in the vicinity of the *MYC* and *PVT1* genes. Individual HPV integration events were also identified at 14q13.2 and 20q11.21 (Fig 2C).

## Comparative genomics

We next examined the overall mutation rate (SNVs + indels) in this NECC cohort in the context of previously published genomic data from SCLC, SCCB, SCC, and ACC (Fig 3A). The median number of coding mutations per tumor was 63.5 (IQR 54.5–99.0) for NECC, 114 (IQR

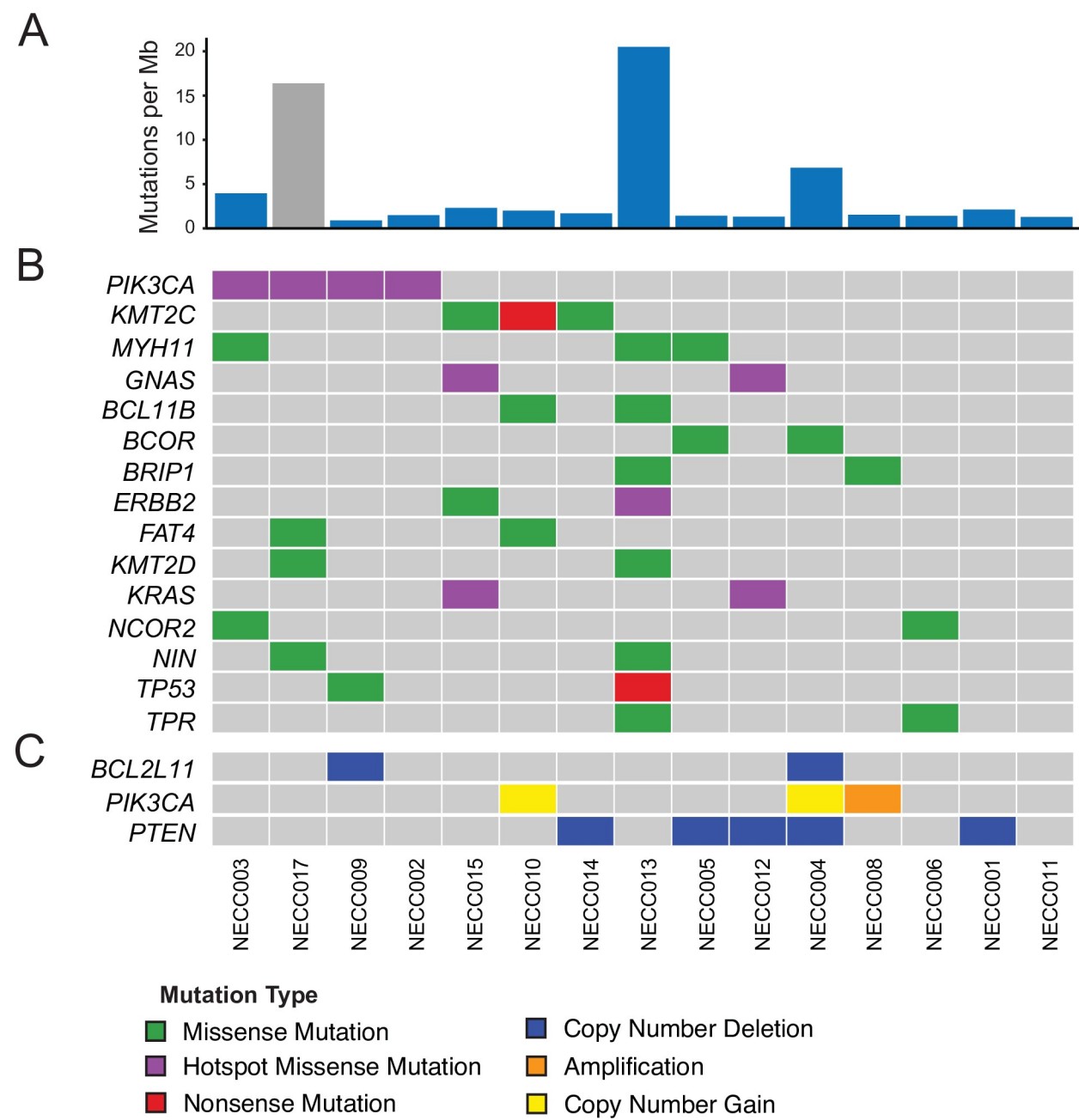

**Fig 1. Somatic mutational landscape of high grade neuroendocrine carcinoma of the cervix. (A)** NECC somatic mutation burden expressed as somatic mutations per Mb. Gray color indicates sample with no available matched normal tissue. **(B)** Genes in COSMIC Cancer Gene Census with recurrent (>1) non-silent somatic mutations. **(C)** Copy number alterations in PI3-kinase pathway members PIK3CA and PTEN, as well as BCL2L11 which was identified as recurrently deleted in NECC (q = .04). NECC = small cell neuroendocrine carcinoma of the cervix, Mb = megabase.

67.3–192.8) for SCC, 83 (IQR 63.8–169.3) for ACC, 261 (IQR 181.5–440.5) for SCCB, and 313 (IQR 235.3–457.3) for SCLC. When compared to NECC, SCLC and SCCB exhibited more coding mutations and this difference was statistically significant (P < .001 for SCLC, P = .001 for SCCB by two-sided Wilcoxon rank-sum test). In contrast, the burden of coding mutations did not differ between NECC and ACC (P = .26) or SCC (P = .053).

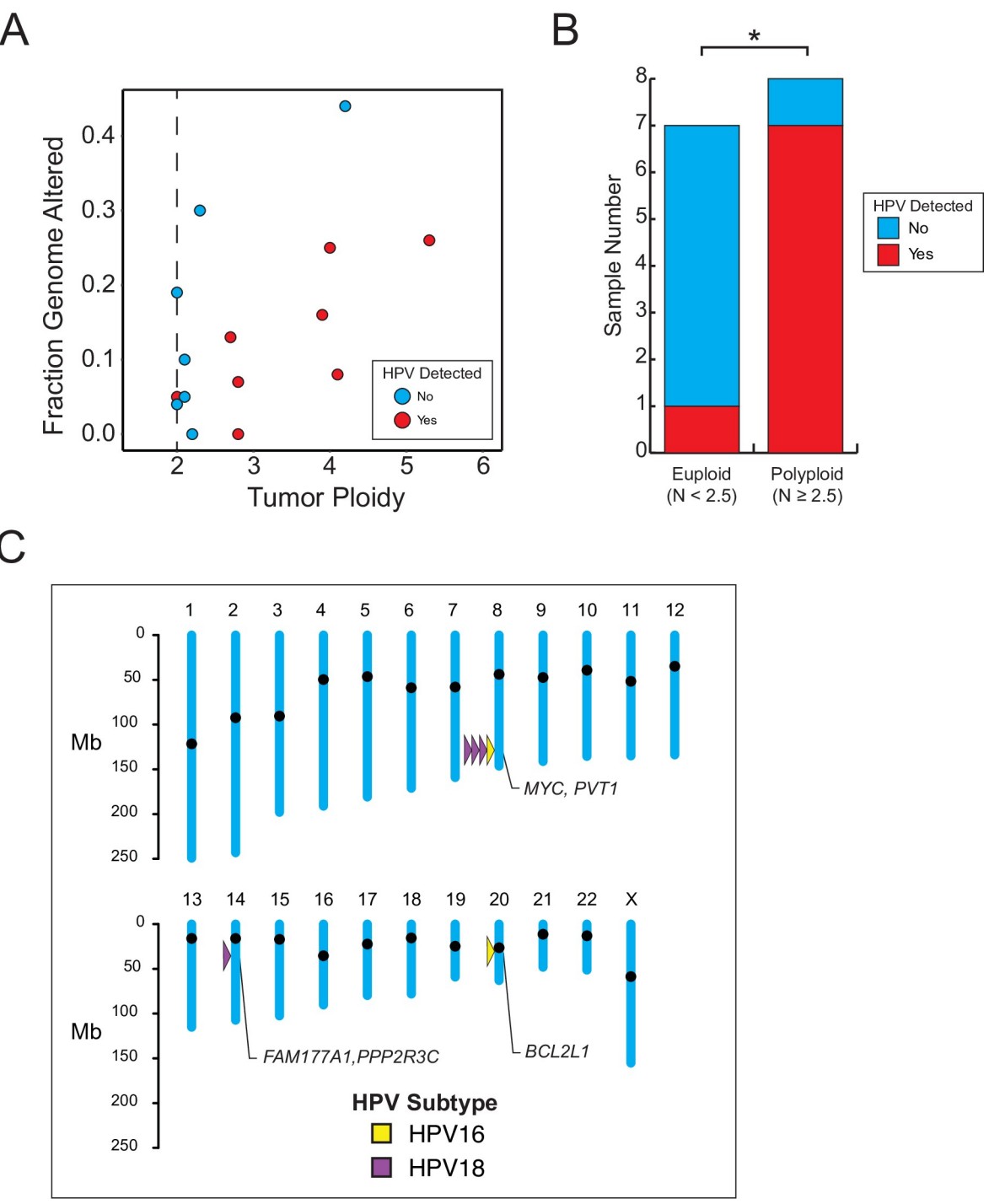

**Fig 2. Gene-adjacent HPV integration events in high grade neuroendocrine carcinoma of the cervix.** (A) Fraction of genome altered by copy number alterations (.3 > log CN ratio > -.3) as a function of tumor ploidy. Red, HPV genome detected; Blue, HPV genome not detected. (B) The presence of HPV is associated with polyploid karyotype in NECC. *, p < .05. (C) Map of gene-adjacent HPV integration events identified in NECC. Yellow, HPV16; Purple, HPV18.

Differences were observed in the frequency of non-silent mutation in specific cancer-related genes between small cell carcinomas of different anatomic origin (Fig 3B). Two NECC

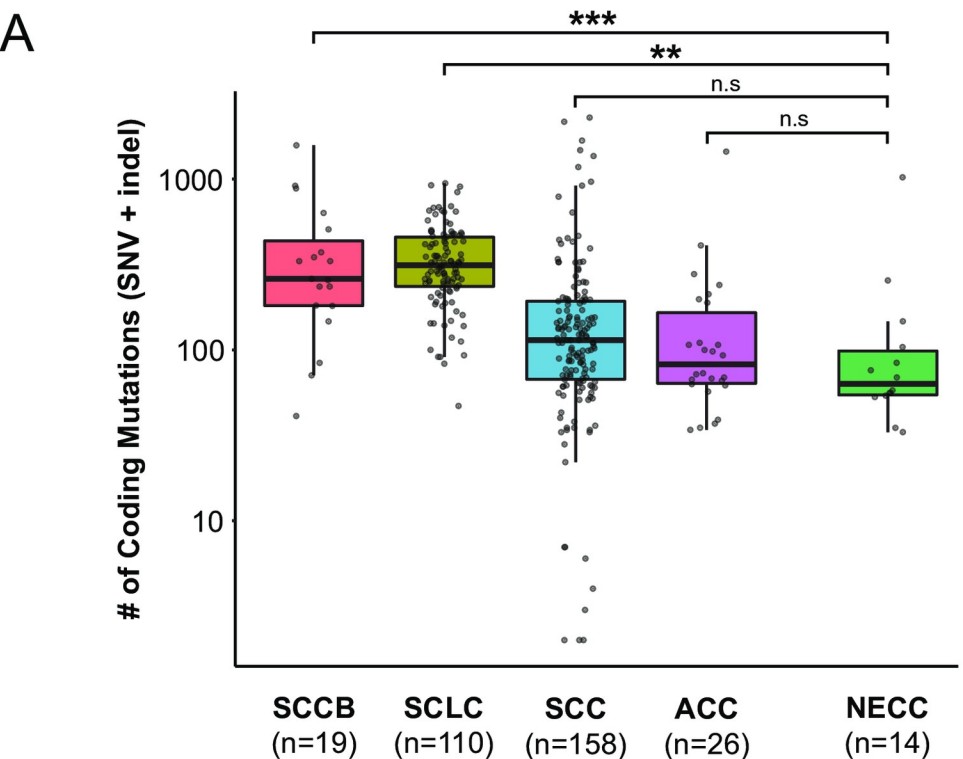

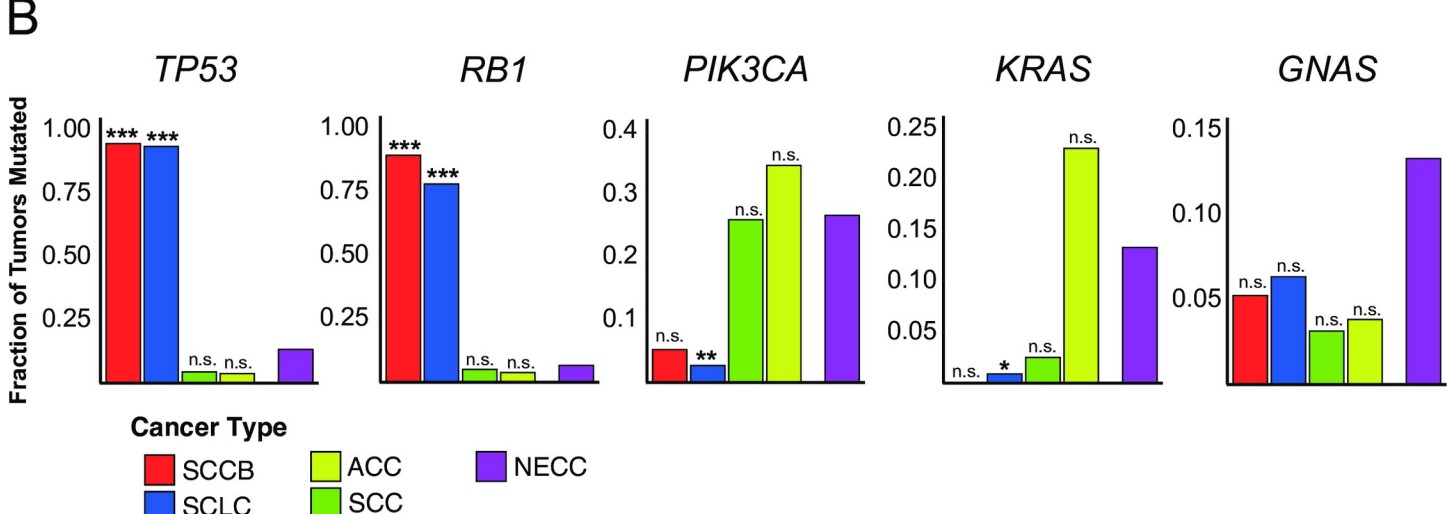

**Fig 3. Comparison of somatic mutation rate among cervical carcinomas and extra-cervical small cell carcinomas of the lung and bladder.** (A) Coding mutation burden by tumor type. Statistical comparisons were performed between NECC and each other tumor type using a two-sided Wilcoxon rank-sum test. (B) Fraction of mutated samples by tumor type for frequently mutated genes. For each gene, statistical comparisons were performed between the mutation rate of NECC and each other tumor type using a two-sided Fisher's exact test. ***, p < .001; **, p < .01; *, p < .05; n.s. = not significant. SNV = single nucleotide variant, indel = small insertion/deletion, SCCB = small cell carcinoma of the bladder, SCLC = small cell lung carcinoma, NECC = small cell neuroendocrine carcinoma of the cervix, SCC = squamous carcinoma of the cervix, ACC = endocervical adenocarcinoma.

tumors (2 of 15 tumors, 13%) had non-silent *TP53* mutations and one had a non-silent *RB1* mutation (1 of 15 tumors, 7%). Compared to NECC, non-silent mutation of *TP53* was significantly more frequent in both SCLC (103 of 110 tumors [94%], P < .001, two-sided Fisher

exact test) and SCCB (18 of 19 tumors [95%], P < .001, two-sided Fisher exact test). Non-silent mutation of *RB1* was similarly more common in SCLC (86 of 110 tumors [78%], P < .001, two-sided Fisher exact test) and SCCB (17 of 19 tumors [89%], P < .001, two-sided Fisher exact test) compared with NECC. No significant difference in *PIK3CA* mutation rate was observed between NECC (4 of 15 tumors [27%]) and either SCC (41 of 158 tumors [26%], P = 1) or ACC (9 of 26 tumors [35%], P = .73). No statistically significant differences were observed between rates of *KRAS* or *GNAS* mutations in NECC compared to other tumor types (all P > 0.05).

Mutational signature decomposition analysis was next used to identify mutational processes acting across tumor types. We found that NECC exhibits contributions from several previously defined mutagenic processes (Fig 4A), including large contributions from an age-related signature related to spontaneous deamination of 5-methylcytosine (Signature 1) and from activation induced cytidine deaminase (AID)/apolipoprotein B editing complex (APOBEC) deaminase activity (COSMIC Signature 2 & Signature 13) [22,23]. We next performed unsupervised hierarchical clustering of tumors based upon the relative signature contribution of each mutational signature. Four clusters were identified which were characterized by a high contribution of defective mismatch repair signature (Cluster 1), age related spontaneous deamination of 5-methylcytosine (Cluster 2), AID/APOBEC cytidine deaminase activity (Cluster 3), and tobacco exposure (Cluster 4) (Fig 4B). SCLC tumors were found almost exclusively in Cluster 4, consistent with prior mutational signature analyses demonstrating a preponderance of tobacco exposure signature in this tumor type (Fig 4C) [19,20]. In contrast, all NECC tumors were found in Cluster 2 or Cluster 3 with a distribution of cluster membership that most closely resembled ACC.

## Discussion

In this study we report the largest series of NECC samples analyzed by WES described to date. Comparative genomics suggests NECC is genetically more similar to common cervical cancer subtypes than to extra-cervical small cell neuroendocrine carcinomas of the lung and bladder. In particular, all three cervix cancer subtypes exhibit frequent evidence of activating PI3-kinase/MAPK pathway mutations and only rarely carry non-silent mutations in the tumor suppressors *TP53* or *RB1*. Among the three cervix cancer subtypes, the total coding mutational burden is similar and a substantial fraction of these mutations are related to AID/APOBEC cytidine deamination mutational processes. APOBEC enzymes have anti-viral activity and have been shown to modify HPV genomes in precancerous cutaneous lesions [27]. Activation of APOBEC enzymes by HPV infection of cervical epithelium cells may in turn lead to an increased rate of somatic mutation accumulation, particularly at TCW motifs (where W corresponds to either A or T) known to be preferred sites of APOBEC-mediated mutagenesis. As has been previously noted, the *PIK3CA* c.1633G>A (p.E545K) mutation occurs at one such APOBEC motif and this association may account for the frequency of this mutation across cervix cancer subtypes [21]. In contrast, we found marked differences between NECC and both SCLC and SCCB. Both types of extra-cervical small cell carcinoma exhibit near universal somatic *TP53* and *RB1* mutation, with mutational burdens in excess of those seen in NECC and distinct mutational signature profiles.

Low frequency mutations were observed among several other genes in NECC. These include the lysine methyltransferase genes *KMT2C* and *KMT2D*, which have been implicated in several other tumor types and suggests that chromatin remodeling activity may play a role in a subset of NECC [28]. *BRIP1* missense mutation was observed in one NECC tumor. Germline mutation of *BRIP1* has been show to increase the risk of hereditary ovarian cancer [29],

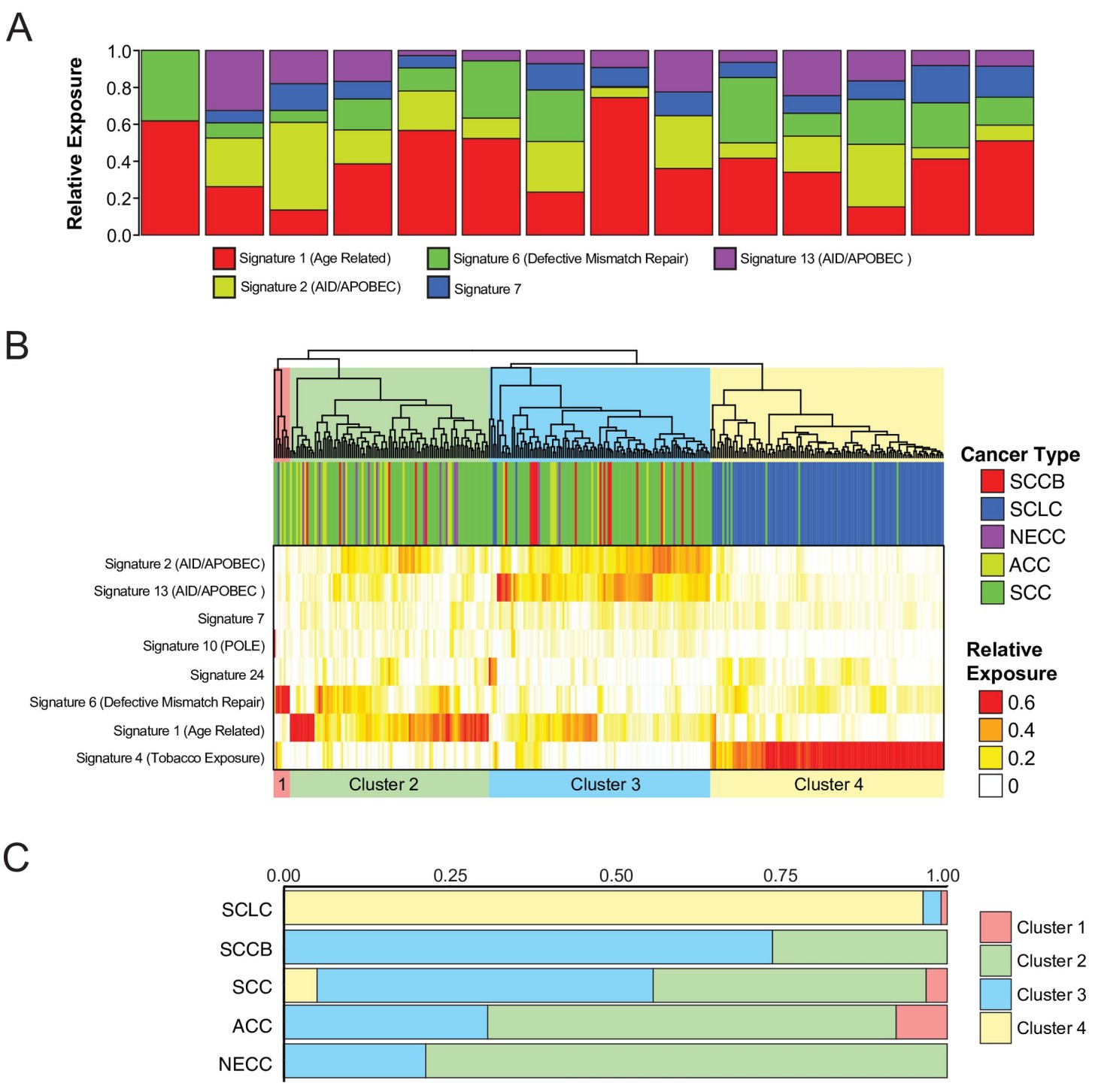

**Fig 4. Mutational signature contribution across tumor types. (A)** SNV mutational signature contribution by NECC sample. **(B)** Unsupervised hierarchical clustering of SNV mutational signature contributions by sample across tumor types. **(C)** Relative cluster membership of sample by tumor type. SNV = single nucleotide variant, SCCB = small cell carcinoma of the bladder, SCLC = small cell lung carcinoma, NECC = small cell neuroendocrine carcinoma of the cervix, SCC = squamous carcinoma of the cervix, ACC = endocervical adenocarcinoma.

although the *BRIP1* NECC mutation we observed was not clearly pathogenic and that tumor did not exhibit mutational signatures consistent with a defect in homologous recombination.

These data thus do not support homologous recombination deficiency as a mechanism of genomic instability in NECC.

HPV is detectable in nearly all SCC and ACC, consistent with the clear oncogenic role played by HPV in these cervical cancer subtypes [21]. A causative relationship between HPV and NECC remains controversial, although a recent meta-analysis of HPV detection in NECC suggest a high prevalence of HPV in this tumor type [30]. In our cohort, we detected a statistically significant association between detectable HPV integration and tumor cell aneuploidy. This finding suggests that HPV may play a role in destabilizing genome integrity and promoting tumor formation in NECC. Interestingly, we identified HPV integration at 8q24.21 in 4 of 15 (27%) of NECC tumor samples analyzed, consistent with previously published rates of HPV integration at this hotspot region in other cervix cancer subtypes [31]. Although 8q24.21 contains the *MYC* oncogene, a causative role for HPV integration at this site in the pathogenesis of cervix cancer has not been established.

This study has several strengths, most notably the relatively large series of a rare tumor type and the use of WES to provide a comprehensive picture of coding mutations, CNVs, and mutational signatures. By using fresh-frozen tissue and matched normal samples, we are able to report high-confidence somatic mutations free from contamination by germline variants or artifacts of formalin fixation. Lastly, these NECC are likely representative of this rare disease since tumor samples used in this study were assigned histopathologic diagnoses at regional cancer referral centers with extensive experience in the diagnosis and treatment of rare gynecologic malignancies. This study is limited by the inability to reliably identify rare recurrent oncogenic mutations in NECC given the small sample size. In addition, we could not comprehensively map HPV integration events across intergenic regions since only coding exons were sequenced.

The challenge of treating patients with rare malignancies is compounded by the frequent absence of data from large, randomized clinical trials to guide the selection of optimal therapies. One useful approach to this dilemma is to turn to data generated from studies of a histologically similar tumor occurring at a distinct anatomic site with much higher prevalence. This approach led to the adoption of platinum-based chemotherapy for the treatment of ovarian dysgerminoma based on data generated among male patients with histologically similar testicular seminomas [32]. More recently, molecular phenotyping has demonstrated that mucinous ovarian carcinomas have similarities with gastrointestinal malignancies, including frequent *KRAS* mutation, and preclinical data suggest this subtype of ovarian cancer may respond to agents such as 5-FU and oxaliplatin used in the treatment of colorectal cancer [33,34].

Small retrospective studies suggest that patients with NECC benefit from combination platinum-based chemotherapy in the upfront setting in NECC [35], and EP chemotherapy is often used in this setting based on clinical trials conducted among SCLC patients. In contrast, patients with advanced or recurrent SCC or ACC are most often treated with a combination of carboplatin, paclitaxel, and bevacizumab based on data from large prospective, randomized clinical trials [36,37]. The genetic similarities we observe between NECC and the more common cervical cancer histologies suggest a re-evaluation of chemotherapeutic approaches to this rare disease may be warranted.

## Supporting information

**S1 Table. Whole exome sequencing summary of high grade neuroendocrine carcinoma of the cervix cohort.**
(XLSX)

## Acknowledgments

An abstract related to this work was presented as a poster at the Society of Gynecologic Oncology (SGO) 23rd Annual Winter Meeting, Feb 08–10, 2018, Snowmass, CO.

## Author Contributions

**Conceptualization:** R. Tyler Hillman, Robert Cardnell, Lauren A. Byers, Mario Leitao, Elizabeth Swisher, Michael Frumovitz.

**Data curation:** R. Tyler Hillman, Robert Cardnell, Junya Fujimoto, Mario Leitao, Elizabeth Swisher, P. Andrew Futreal, Michael Frumovitz.

**Formal analysis:** R. Tyler Hillman, Junya Fujimoto, Won-Chul Lee, Jianjun Zhang, Lauren A. Byers, Preetha Ramalingam, P. Andrew Futreal.

**Funding acquisition:** P. Andrew Futreal, Michael Frumovitz.

**Investigation:** R. Tyler Hillman, Robert Cardnell, Mario Leitao, P. Andrew Futreal, Michael Frumovitz.

**Methodology:** Elizabeth Swisher, Michael Frumovitz.

**Project administration:** P. Andrew Futreal, Michael Frumovitz.

**Resources:** P. Andrew Futreal, Michael Frumovitz.

**Supervision:** Jianjun Zhang, Lauren A. Byers, Preetha Ramalingam, Mario Leitao, Elizabeth Swisher, P. Andrew Futreal, Michael Frumovitz.

**Writing – original draft:** R. Tyler Hillman, Michael Frumovitz.

**Writing – review & editing:** R. Tyler Hillman, Robert Cardnell, Junya Fujimoto, Won-Chul Lee, Jianjun Zhang, Lauren A. Byers, Preetha Ramalingam, Mario Leitao, Elizabeth Swisher, P. Andrew Futreal, Michael Frumovitz.

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
