## [Decision Letter · Decision Letter 0]

13 Mar 2020

PONE-D-20-05313

Comparative genomics of high grade neuroendocrine carcinoma of the cervix

PLOS ONE

Dear Authors,

Thank you for submitting your manuscript to PLOS ONE. After careful consideration, we feel that it has merit but does not fully meet PLOS ONE’s publication criteria as it currently stands. Therefore, we invite you to submit a revised version of the manuscript that addresses the points raised during the review process.

Please follow reviewers' comments to revise your manuscript. In particular, is the study conclusive with such a small sample size? Are there other ways to validate the results?

We would appreciate receiving your revised manuscript by Apr 27 2020 11:59PM. To enhance the reproducibility of your results, we recommend that if applicable you deposit your laboratory protocols in protocols.io, where a protocol can be assigned its own identifier (DOI) such that it can be cited independently in the future. For instructions see: http://journals.plos.org/plosone/s/submission-guidelines#loc-laboratory-protocols

We look forward to receiving your revised manuscript.

Kind regards,

Jingwu Xie

Academic Editor

PLOS ONE

Journal Requirements:

Additional Editor Comments (if provided):

Please follow reviewer's comments to revise your manuscript. In particular, is the study conclusive with such a small sample? Are there other ways to validate the findings?

Reviewers' comments:

Reviewer's Responses to Questions

**Comments to the Author**

1. Is the manuscript technically sound, and do the data support the conclusions?

Reviewer #1: Partly

Reviewer #2: Yes

2. Has the statistical analysis been performed appropriately and rigorously? 

Reviewer #1: Yes

Reviewer #2: Yes

3. Have the authors made all data underlying the findings in their manuscript fully available?

Reviewer #1: Yes

Reviewer #2: Yes

4. Is the manuscript presented in an intelligible fashion and written in standard English?

Reviewer #1: Yes

Reviewer #2: Yes

5. Review Comments to the Author

Reviewer #1: Please find below my review of the manuscript entitled “Comparative genomics of high-grade neuroendocrine carcinoma of the cervix” submitted by Robert Tyler Hillman et al. The manuscript No. is PONE-D-20-05313.

Robert Tyler Hillman and other co-authors intended to comprehensively investigate the genomics landscape of high-grade neuroendocrine carcinoma of the cervix (NECC) through multiple algorithms of computational biology. NECC is a rare type of malignant cervical cancers with worse overall survival compared to other types of cervical cancer. Development of an effective treatment is critical to improve patients’ life quality. The study is meaningful, but the less support for its clinical translational significance.

The topic of this manuscript has potential for a clinical translational apply, but the quality of the studies in it looks a little bit modest and superficial. In the spirit of making the work better, I sincerely provide my advices to the authors.

First, whole exome sequencing and multiple bioinformatics analysis were used to identify the genomics of NECC in this manuscript. The identified mutations in NECC represented the genomic changes under tumor condition, these may have implications in NECC development and therapy response, a deep literature review and discussion will make a more clarity of the Importance of the mutations.

Second, the authors identified and compared mutations and CNVs in NECC and matched normal tissue, the sample size is quite small, the frequency of some mutations was relatively low, the comparison may be invalid. Meanwhile, comparative analysis was performed to explore the difference between NECC and other types of cervical cancers and extra-cervical small cell carcinomas, the clinical stage should be moderately matched.

Third, as the authors declared, the sample size is low, and the clinical implication was limited, although the authors try to extend the results to suggest some potential therapies that patients may benefit. Further validation should be conducted.

In addition, the abstract could be more summative and concise.

In summary, the bioinformatics analysis is ok, but the sample size is relatively small, so the clinical translational significance of the findings may be limited. If a bigger sample size and more clinical data support for the conclusion, the paper will be good to accept.

Reviewer #2: The authors performed the whole exome sequencing for 15 high grade neuroendocrine carcinoma of the cervix (NECC) samples and matched normals (one common normal). They found 67% of NECC had PI3K or MAPK pathway activation mutations and 13% and 7% of NECC had TP53 and RB1 mutations respectively. When compared with NECC, small cell lung cancer (SCLC) and small cell carcinoma of the bladder (SCCB) had quite different mutation patterns with high frequency in TP53 and RB1 mutations, and relatively low frequency in PIK3CA and KRAS mutations. In contrast, the other two types of cervical cancer, squamous carcinoma of the cervix (SCC) and endocervical adenocarcinoma (ACC), had the similar mutation patterns with the NECC. Based on their results and the current chemotherapy strategy of NECC, the authors suggested to re-evaluate the chemotherapeutic approaches to this rare disease.

There are some questions needed revision.

1 Add the subtitles in the Results section.

2 Please explain the reason why the number of NECC used in analysis is 14 not 15 in Figure 3A

3 The results of PIK3CA and KRAS in Figure 3B did not mentioned in the Results section. They also supported the results that SCC and ACC had the similar mutation patterns with NECC

4 Figure 4C did not cited in the Results.

5 Why did the authors choose GISTIC2.0 to perform the CNV analysis, the software is not commonly used in the field.

6 The authors performed analysis on SNV and CNV, can these results support the conclusion” Comparative genomics suggests NECC is genetically more similar to common cervical cancer subtypes than to extra-cervical small cell neuroendocrine carcinomas of the lung and bladder.”?

6. PLOS authors have the option to publish the peer review history of their article (what does this mean?). If published, this will include your full peer review and any attached files.

Reviewer #1: No

Reviewer #2: No

---

## [Author Response · Author response to Decision Letter 0]

8 Apr 2020

5. Review Comments to the Author

Reviewer #1: Please find below my review of the manuscript entitled “Comparative genomics of high-grade neuroendocrine carcinoma of the cervix” submitted by Robert Tyler Hillman et al. The manuscript No. is PONE-D-20-05313.

Robert Tyler Hillman and other co-authors intended to comprehensively investigate the genomics landscape of high-grade neuroendocrine carcinoma of the cervix (NECC) through multiple algorithms of computational biology. NECC is a rare type of malignant cervical cancers with worse overall survival compared to other types of cervical cancer. Development of an effective treatment is critical to improve patients’ life quality. The study is meaningful, but the less support for its clinical translational significance.

The topic of this manuscript has potential for a clinical translational apply, but the quality of the studies in it looks a little bit modest and superficial. In the spirit of making the work better, I sincerely provide my advices to the authors.

First, whole exome sequencing and multiple bioinformatics analysis were used to identify the genomics of NECC in this manuscript. The identified mutations in NECC represented the genomic changes under tumor condition, these may have implications in NECC development and therapy response, a deep literature review and discussion will make a more clarity of the Importance of the mutations.

We have substantially expanded the Discussion section to accommodate additional literature review and discussion of several other rare mutations observed in a subset of NECC (lines 404-412). We remain careful about ascribing functional significance to a mutation observed in only one tumor sample (e.g. BRIP1) since it is not possible to determine with certainty if this represents a passenger mutation or a true driver event. Nevertheless, we agree with the Reviewer that further discussion of these less common mutations was warranted and have revised the manuscript accordingly.

Second, the authors identified and compared mutations and CNVs in NECC and matched normal tissue, the sample size is quite small, the frequency of some mutations was relatively low, the comparison may be invalid. Meanwhile, comparative analysis was performed to explore the difference between NECC and other types of cervical cancers and extra-cervical small cell carcinomas, the clinical stage should be moderately matched.

We agree with the reviewer that clinical stage should be moderately matched across the cervical cancer cohorts. This is indeed the case in our cohorts, with 77% of adenocarcinoma/squamous cell carcinomas being local stage I-II tumors compared to 73% of the NECC cohort. We have updated the manuscript to include a description of the clinical stages in these cohorts [lines 208-210].

Third, as the authors declared, the sample size is low, and the clinical implication was limited, although the authors try to extend the results to suggest some potential therapies that patients may benefit. Further validation should be conducted.

 NECC is an extraordinarily rare disease, with only approximately 100 to 200 new cases diagnosed each year in the entire United States. Moreover, the analyses described in this manuscript require fresh frozen tissue along with matched normal somatic tissue – sample types that are not collected under routine clinical practice and thus cannot be obtained from pathology archives. The current study represents a close collaboration between three high-volume referral centers (MD Anderson Cancer Center, Memorial Sloan Kettering Cancer Center, and the University of Washington), which pooled all available NECC samples. Given these limitations, is therefore not feasible to analyze a true independent validation cohort because none exists.

 In the published literature to date only 39 NECC have ever been analyzed by any NGS platform: 24 by 50-gene panel sequencing [1], 10 by ~600-gene panel sequencing [2], and 5 by WES of formalin-fixed paraffin-embedded tissue [3]. In this context, our analysis of 15 fresh frozen NECC samples and matched normal tissue by WES represents a substantial contribution to our understanding of this rare disease – increasing the total number of NECC analyzed by WES to date by a factor of 4. 

In the absence of an available external validation cohort, similarities between our findings and prior work in NECC suggests that our results are representative of this rare tumor type. For example, we identify PIK3CA activating mutations in 27% of tumors, compared to 18% in Frumovitz et al [1] and 30% in Xing et al [2]. Rates of activating MAPK mutations and TP53 mutations are also similar among the cohorts. Thus our results are broadly consistent with these prior efforts in areas where they can be directly compared, and these similarities serve to give validity and added confidence to the other findings we report.

In addition, the abstract could be more summative and concise.

We have substantially revised the abstract to make it more summative and concise, and to conform more closely with the abstract style typically used in this journal.

In summary, the bioinformatics analysis is ok, but the sample size is relatively small, so the clinical translational significance of the findings may be limited. If a bigger sample size and more clinical data support for the conclusion, the paper will be good to accept.

We agree with the Reviewer that it would be very interesting to have clinical outcomes and treatment histories for all patients included in this NECC cohort. However, these data were not collected under the collaborative data sharing agreement between the three institutions (MD Anderson Cancer Center, Memorial Sloan Kettering Cancer Center, and the University of Washington). A detailed examination of clinical outcomes stratified by genomic biomarkers would be a very interesting follow up study, but falls outside the scope of the present work which aimed to present comparative genomics between NECC, more common cervical cancer subtypes, and small cell carcinomas of the lung and bladder. Please see above regarding why it is not feasible to increase the sample size, given the extreme rarity of this tumor type. 

Reviewer #2: The authors performed the whole exome sequencing for 15 high grade neuroendocrine carcinoma of the cervix (NECC) samples and matched normals (one common normal). They found 67% of NECC had PI3K or MAPK pathway activation mutations and 13% and 7% of NECC had TP53 and RB1 mutations respectively. When compared with NECC, small cell lung cancer (SCLC) and small cell carcinoma of the bladder (SCCB) had quite different mutation patterns with high frequency in TP53 and RB1 mutations, and relatively low frequency in PIK3CA and KRAS mutations. In contrast, the other two types of cervical cancer, squamous carcinoma of the cervix (SCC) and endocervical adenocarcinoma (ACC), had the similar mutation patterns with the NECC. Based on their results and the current chemotherapy strategy of NECC, the authors suggested to re-evaluate the chemotherapeutic approaches to this rare disease.

There are some questions needed revision.

1 Add the subtitles in the Results section.

We have now added subtitles/sub-headings to the Results section and we agree with the reviewer that these add significant clarity.

2 Please explain the reason why the number of NECC used in analysis is 14 not 15 in Figure 3A

Sample NECC017 did not have matched normal tissue for use in subtracting germline variants from WES data. Thus the rate of somatic mutation cannot be accurately calculated for this sample and it was not included in the comparative analysis of mutation rates across tumor types, as presented in Figure 3A.

3 The results of PIK3CA and KRAS in Figure 3B did not mentioned in the Results section. They also supported the results that SCC and ACC had the similar mutation patterns with NECC

We have added statistical comparisons to the manuscript text between PIK3CA, KRAS, and GNAS mutations rates in NECC and other tumor types [lines 292-295]. We agree with the Reviewer that this adds clarity to the text and supports the overall argument of the manuscript that NECC are genetically similar to SCC and ACC.

4 Figure 4C did not cited in the Results.

We thank the Reviewer for detecting this omission, which has now been corrected (line 308).

5 Why did the authors choose GISTIC2.0 to perform the CNV analysis, the software is not commonly used in the field.

We chose to use GISTIC2.0 for CNV analysis because it is the most updated version of a the widely used GISTIC algorithm that has become a standard in the field for CNV analysis. The two papers that describe this algorithm have collectively been cited more than 2000 times [4,5]. Moreover GISTIC or GISTIC2.0 have been used in numerous publications by The Cancer Genome Atlas consortium, including those projects analyzing prostate cancer [6], ovarian cancer [7], esophageal cancer [8], endometrial cancer [9], cervical cancer [10], and lung cancer [11] to name a few.

6 The authors performed analysis on SNV and CNV, can these results support the conclusion” Comparative genomics suggests NECC is genetically more similar to common cervical cancer 

subtypes than to extra-cervical small cell neuroendocrine carcinomas of the lung and bladder.”?

 We agree with the Reviewer that the data presented in this manuscript does indeed support this conclusion, and we have added a similar statement to the Discussion section to emphasize this point (lines 314-316).

REFERENCES

[1] M. Frumovitz, J.K. Burzawa, L.A. Byers, Y.A. Lyons, P. Ramalingam, R.L. Coleman, et al., Sequencing of mutational hotspots in cancer-related genes in small cell neuroendocrine cervical cancer, Gynecol. Oncol. 141 (2016) 588–591. doi:10.1016/j.ygyno.2016.04.001.

[2] D. Xing, G. Zheng, J.K. Schoolmeester, Z. Li, A. Pallavajjala, L. Haley, et al., Next-generation Sequencing Reveals Recurrent Somatic Mutations in Small Cell Neuroendocrine Carcinoma of the Uterine Cervix., Am. J. Surg. Pathol. 42 (2018) 750–760. doi:10.1097/PAS.0000000000001042.

[3] S.Y. Cho, M. Choi, H.J. Ban, C.H. Lee, S. Park, H. Kim, et al., Cervical small cell neuroendocrine tumor mutation profiles via whole exome sequencing, Oncotarget. 8 (2017) 8095–8104. doi:10.18632/oncotarget.14098.

[4] C.H. Mermel, S.E. Schumacher, B. Hill, M.L. Meyerson, R. Beroukhim, G. Getz, GISTIC2.0 facilitates sensitive and confident localization of the targets of focal somatic copy-number alteration in human cancers, Genome Biol. 12 (2011) R41. doi:10.1186/gb-2011-12-4-r41.

[5] R. Beroukhim, G. Getz, L. Nghiemphu, J. Barretina, T. Hsueh, D. Linhart, et al., Assessing the significance of chromosomal aberrations in cancer: methodology and application to glioma., Proc. Natl. Acad. Sci. U. S. A. 104 (2007) 20007–12. doi:10.1073/pnas.0710052104.

[6] A. Abeshouse, J. Ahn, R. Akbani, A. Ally, S. Amin, C.D. Andry, et al., The Molecular Taxonomy of Primary Prostate Cancer, Cell. 163 (2015) 1011–1025. doi:10.1016/j.cell.2015.10.025.

[7] The Cancer Genome Atlas, Integrated genomic analyses of ovarian carcinoma., Nature. 474 (2011) 609–615. doi:10.1038/nature10166.

[8] J. Kim, R. Bowlby, A.J. Mungall, A.G. Robertson, R.D. Odze, A.D. Cherniack, et al., Integrated genomic characterization of oesophageal carcinoma, Nature. (2017). doi:10.1038/nature20805.

[9] C. genome A.R. Network, Integrated genomic characterization of endometrial carcinoma., Nature. 497 (2013) 67–73. doi:10.1038/nature12113.

[10] R.D. Burk, Z. Chen, C. Saller, K. Tarvin, A.L. Carvalho, C. Scapulatempo-Neto, et al., Integrated genomic and molecular characterization of cervical cancer, Nature. 228 (2017). doi:10.1038/nature21386.

[11] T.C.G.A.R. Network, (Participants are arranged by area of contribution and then by institution.), G. sequencing centres: B. Institute, P.S. Hammerman, M.S. Lawrence, D. Voet, et al., Comprehensive genomic characterization of squamous cell lung cancers., Nature. 489 (2012) 519–525. doi:10.1038/nature11404.

---

## [Decision Letter · Decision Letter 1]

28 May 2020

Comparative genomics of high grade neuroendocrine carcinoma of the cervix

PONE-D-20-05313R1

Dear Dr. Hillman,

We are pleased to inform you that your manuscript has been judged scientifically suitable for publication and will be formally accepted for publication once it complies with all outstanding technical requirements.

With kind regards,

Hari K Koul, Ph.D., FACN,FASN

Academic Editor

PLOS ONE

Additional Editor Comments (optional):

Reviewers' comments:

Reviewer's Responses to Questions

**Comments to the Author**

1. If the authors have adequately addressed your comments raised in a previous round of review and you feel that this manuscript is now acceptable for publication, you may indicate that here to bypass the “Comments to the Author” section, enter your conflict of interest statement in the “Confidential to Editor” section, and submit your "Accept" recommendation.

Reviewer #1: All comments have been addressed

Reviewer #2: All comments have been addressed

2. Is the manuscript technically sound, and do the data support the conclusions?

Reviewer #1: Yes

Reviewer #2: Yes

3. Has the statistical analysis been performed appropriately and rigorously? 

Reviewer #1: Yes

Reviewer #2: Yes

4. Have the authors made all data underlying the findings in their manuscript fully available?

Reviewer #1: Yes

Reviewer #2: Yes

5. Is the manuscript presented in an intelligible fashion and written in standard English?

Reviewer #1: Yes

Reviewer #2: Yes

6. Review Comments to the Author

Reviewer #1: Dear Editor,

Please find below my re-review of the manuscript entitled “Comparative genomics of high-grade neuroendocrine carcinoma of the cervix” submitted by Robert Tyler Hillman et al. The manuscript No. is PONE-D-20-05313R1.

Robert Tyler Hillman and other co-authors intended to comprehensively investigate the genomics landscape of high-grade neuroendocrine carcinoma of the cervix (NECC) through multiple algorithms of computational biology. As the author declared, NECC is a very rare type of malignant cervical cancer, which is hence extremely difficult to collect a big sample composed cohort for the analysis. After a careful consideration on this limitation, the review accepted the authors' explain. Also, the authors have adequately addressed my other comments raised in a previous round of review, and looks that this manuscript is acceptable for publication now. So I sincerely hope this simple but interesting work could be published on PLOS ONE in the near future.

Please feel free to contact me if you require further details.

Reviewer #2: (No Response)

7. PLOS authors have the option to publish the peer review history of their article (what does this mean?). If published, this will include your full peer review and any attached files.

Reviewer #1: No

Reviewer #2: No

---

## [Editor Report · Acceptance letter]

29 May 2020

PONE-D-20-05313R1 

Comparative genomics of high grade neuroendocrine carcinoma of the cervix 

Dear Dr. Hillman:

I am pleased to inform you that your manuscript has been deemed suitable for publication in PLOS ONE. Congratulations! Your manuscript is now with our production department. 

With kind regards,

on behalf of

Prof (Dr) Hari K Koul 

Academic Editor

PLOS ONE